# Insights into the Value of Lyso-Gb1 as a Predictive Biomarker in Treatment-Naïve Patients with Gaucher Disease Type 1 in the LYSO-PROOF Study

**DOI:** 10.3390/diagnostics13172812

**Published:** 2023-08-30

**Authors:** Filipa Curado, Sabine Rösner, Susanne Zielke, Gina Westphal, Ulrike Grittner, Volha Skrahina, Mohammed Alasel, Ahmad Mehmood Malik, Christian Beetz, Tobias Böttcher, Gal Barel, Ashish Prasad Sah, Tama Dinur, Nadeem Anjum, Quidad Ichraf, Yamna Kriouile, Zahra Hadipour, Fatemeh Hadipour, Shoshana Revel-Vilk, Claudia Cozma, Jörg Hartkamp, Huma Cheema, Ari Zimran, Peter Bauer, Arndt Rolfs

**Affiliations:** 1CENTOGENE GmbH, 18055 Rostock, Germany; filipa.curado@centogene.com (F.C.); sabine.roesner@centogene.com (S.R.); susanne.zielke@mail.de (S.Z.); gina.westphal@centogene.com (G.W.); alasel.mohammed@gmail.com (M.A.); ahmad_m.malik@outlook.com (A.M.M.); christian.beetz@centogene.com (C.B.); tobias.boettcher@centogene.com (T.B.); ashish.sah@centogene.com (A.P.S.); claudia.cozma@centogene.com (C.C.); jhartkamp@gmx.de (J.H.); 2Berlin Institute of Health, Charité-Universitätsmedizin Berlin, 10117 Berlin, Germany; ulrike.grittner@charite.de; 3Institute of Biometry and Clinical Epidemiology, Charité-Universitätsmedizin Berlin, 10117 Berlin, Germany; 4Rhythm Pharmaceuticals Inc., 20095 Hamburg, Germany; skrahinavolha@gmail.com; 5Gaucher Unit, Shaare Zedek Medical Center, Jerusalem 9103102, Israel; dinurtama@gmail.com (T.D.); srevelvilk@gmail.com (S.R.-V.); azimran@gmail.com (A.Z.); 6The Children’s Hospital and University of Child Health Sciences, Lahore 54600, Pakistan; nadeemanjum1900@yahoo.com (N.A.); pedgilahore@gmail.com (H.C.); 7Children Hospital’s Rabat, Neuropediatric-Metabolic, Rabat 6527, Morocco; wichraf1990@gmail.com (Q.I.); dr.kriouile@gmail.com (Y.K.); 8Soodbakhash Poly Clinic, Atiyeh Hospital, Tehran 1416753955, Iran; dr.hadipour@yahoo.com (Z.H.); f.ashuo@yahoo.com (F.H.); 9Medical Genetics Department, Pars Research Center & Hospital, Tehran 1416753955, Iran; 10Faculty of Medicine, Hebrew University of Jerusalem, Jerusalem 9112002, Israel; 11Medical Faculty, University of Rostock, 18057 Rostock, Germany; arndt.rolfs@med.uni-rostock.de; 12Agyany Pharmaceutics Ltd., Jerusalem 9103102, Israel; 13RCV Rare Disease GmbH, 10115 Berlin, Germany

**Keywords:** Gaucher disease, lyso-Gb1, glucosylsphingosine, biomarkers, GD-DS3

## Abstract

Gaucher disease (GD) is a rare autosomal recessive disorder arising from bi-allelic variants in the *GBA1* gene, encoding glucocerebrosidase. Deficiency of this enzyme leads to progressive accumulation of the sphingolipid glucosylsphingosine (lyso-Gb1). The international, multicenter, observational “Lyso-Gb1 as a Long-term Prognostic Biomarker in Gaucher Disease”—LYSO-PROOF study succeeded in enrolling a cohort of 160 treatment-naïve GD patients from diverse geographic regions and evaluated the potential of lyso-Gb1 as a specific biomarker for GD. Using genotypes based on established classifications for clinical presentation, patients were stratified into type 1 GD (*n* = 114) and further subdivided into mild (*n* = 66) and severe type 1 GD (*n* = 48). Due to having previously unreported genotypes, 46 patients could not be classified. Though lyso-Gb1 values at enrollment were widely distributed, they displayed a moderate and statistically highly significant correlation with disease severity measured by the GD-DS3 scoring system in all GD patients (r = 0.602, *p* < 0.0001). These findings support the utility of lyso-Gb1 as a sensitive biomarker for GD and indicate that it could help to predict the clinical course of patients with undescribed genotypes to improve personalized care in the future.

## 1. Introduction

Gaucher disease (GD) is one of the most common inherited lysosomal storage disorders (LSDs) and is caused by bi-allelic variants in the beta-glucocerebrosidase *(GBA1)* gene [1]. Importantly, diminished activity of this enzyme leads to the accumulation of glucocerebroside as well as with other glycolipids in the cells of the monocyte–macrophage system, causing a multisystemic disease with huge phenotypic heterogeneity [2]. The estimated prevalence of GD ranges from 1:50,000 to 1:100,000 in the general population; whereas, in the Ashkenazi Jewish population, ~1:855 are affected [3].

GD represents a wide spectrum of phenotypes from a perinatal-lethal to an asymptomatic form and is typically subdivided into three main types. The non-neuronopathic type 1 GD (GD1) is the most prevalent subclass among Caucasians and is most often characterized by presence of bone disease, organomegaly (liver and spleen), hematologic abnormalities (anemia and thrombocytopenia), and a lack of primary central nervous system involvement [2]. Acute neuronopathic type 2 GD (GD2) and subacute neuronopathic type 3 GD (GD3) are both characterized by the additional presence of primary neurologic disease. The prevalence of neuropathic GD (nGD) varies across ethnic groups but appears to be higher among those who are not of European origin. Although these classifications have clinical utility, these phenotypes are a continuum, as common for all other LSDs [4].

To more precisely measure and assess clinical symptoms and disease severity (burden of disease) in patients with GD1, a disease severity scoring system (GD-DS3) has been introduced [5,6]. GD-DS3 is a validated measure for GD1 disease severity based on bone, hematologic, and visceral domains, which can be used to monitor long-term outcomes of patients starting treatment [7].

The identification of an ideal biomarker for GD, aiding in diagnosis, prognosis, monitoring, and pathophysiologic understanding, has proven difficult due to the complexity of this monogenetic disease [8,9]. It has been suggested that the clinical and biochemical heterogeneity of GD might be partially explained by modifier genes, epigenetics, and external factors; however, their potential impact is currently largely unknown [10].

In the past, plasma biomarkers, such as alkaline phosphatase, angiotensin-converting enzyme (ACE), ferritin, and high-density lipoprotein, have been utilized; however, they are influenced by other factors and are not all specific for GD [11,12]. Similarly, the more widely used plasma biomarkers hydrolase chitotriosidase and chemokine CCL18 are both not central to disease pathophysiology and not specific to GD [13,14]. Additionally, approximately 6% of the population is deficient in chitotriosidase activity due to benign variants in the *CHIT1* gene [15]. In search for highly specific GD biomarkers, two groups independently identified glucosylsphingosine (lyso-Gb1), the deacetylated form of glucosylceramide, and confirmed its high specificity and sensitivity for the diagnosis of GD [16,17]. Subsequently, lyso-Gb1 levels were shown to decrease upon response to treatment [18,19,20,21,22]—strongly supporting lyso-Gb1 as a disease monitoring biomarker for GD.

Dried blood spot (DBS)-based quantification of lyso-Gb1 has previously been established as a feasible and highly valid strategy for therapeutic monitoring of GD patients [23]. The use of DBS for diagnosis has many advantages, including (a) easy collection, (b) only a small amount of blood is needed, and (c) samples can be sent via regular mail at room temperature. As both lyso-Gb1 levels and molecular analysis can be performed accurately from the same small sample volume, diagnosis of GD based on lyso-Gb1 measurements in combination with confirmatory *GBA1* mutation analyses in DBS has been recommended to become the new standard for screening of patients suspected of GD [24].

Using the clinical, genetic, and biochemical data from the 160 newly diagnosed treatment-naïve GD patients from the “Lyso-Gb1 as a Long-term Prognostic Biomarker in Gaucher Disease” (LYSO-PROOF) study, we aimed to investigate the application of lyso-Gb1 as a predictive biomarker for the clinical severity of the patients’ genotype. 

## 2. Materials and Methods

All DBS samples were analyzed at CENTOGENE GmbH, Rostock, Germany. Genotyping was performed via whole *GBA1* sequencing (exons and exon–intron boundaries) from DNA extracted from all DBS samples independent of the lyso-Gb1 levels. The sequencing approach would have detected most types of *GBA1* aberrations, including *GBAP*-mediated recombination events. The definitions of mild versus severe GD1 genotypes follow: c.1226A > G p.(Asn409Ser) homozygous was categorized as “mild” and all other GD1 genotypes as “severe”. Testing of lyso-Gb1 was performed using the collected DBS samples; the lyso-Gb1 levels were then analyzed according to the previously described method [24]. 

### 2.1. Sample Analysis

Sample collection: DBS cards were prepared by dropping 60 µL of blood on CentoCard^®^ filter cards (CENTOGENE GmbH, Rostock, Germany). The blood was then left to dry for 2–4 h at room temperature. Before analysis, 3.2 mm discs were cut out from the homogenous parts of the DBS, using a DBS puncher (Perkin Elmer LAS GmbH, Hamburg, Germany), with each disc containing approximately 3.1 µL blood. 

Sample preparation: For each participant, 3 DBS discs were cut out and transferred into a round-bottom 2 mL tube (Sarstedt AG & Co. KG, Nümbrecht, Germany) and 50 µL extraction solution (DMSO/water, 1/1) together with 100 µL internal standard solution (200 ng/mL Lyso-Gb2 (Biotrend Chemikalien GmbH, Köln, Germany)) in ethanol were added to the tube and incubated for 30 min, at 37 °C, under agitation at 700 rpm. Samples were briefly sonicated for 1 min before they were transferred to a PALL-8048 96-well filter plate with a PTFE membrane (WVR International GmbH, Dresden, Germany) on top of a 96-well V-shaped plate (WVR International GmbH, Dresden, Germany). Cellular and paper debris were filtered by centrifugation for 5 min, at 3500 rpm, in a Hermle Z300 plate centrifuge (Hermle Labortechnik GmbH, Wehningen, Germany). The V-shaped plate was covered with aluminum foil and inserted into the sample manager.

LC/MS Method: The collected material from the DBS cards was briefly separated by liquid chromatography on an ACE 3 C8, 50 × 2.1 mm column (MZ-Analysentechnik GmbH, Mainz, Germany) using a Waters I-Class UPLC (Waters GmbH, Eschborn, Germany). Solvents: 50 mM FA in water (A) and 50 mM FA in acetone/acetonitrile, 1/1 (B). A flow rate of 0.9 mL/min preheated at 60 °C was used. The gradient was linear, and the analytes were eluted between 40% and 70% solvent B. The UPLC was coupled with an AB-Sciex TQ-5500 (AB Sciex Germany GmbH, Darmstadt, Germany) mass spectrometer, using a 3:1 splitter. An MRM method for monitoring the analytes with the following settings was used: curtain gas, 40 psi; ion spray, 5500 V; desolvation temperature, 500 °C; declustering potential, 40 V; entrance potential, 10 V; collision energy, 30 V; and MRM transitions, lyso-Gb1 (462.3–282.2), and lyso-Gb2 internal standard (624.3–282.2).

Quantification: A 7-point calibration line was added to each plate before measurement. Preparation of the calibration line was analogous to that of the samples with the exception of the DBSs being replaced with standard solutions of increasing Lyso-Gb1 (Biotrend Chemikalien GmbH, Köln, Germany) concentrations: 0, 1, 5, 10, 50, 100, and 200 ng/mL. Analysis and quantification were performed by using Analyst 1.6.2 software (AB Sciex Germany GmbH, Darmstadt, Germany).

### 2.2. Statistical Analysis

To report summary descriptive statistics, we used mean and standard deviation (SD) or median, interquartile range [IQR] for continuous variables, depending on the distribution. Additionally, we report the range of values as minimum and maximum. The Kruskal–Wallis H test was used to determine statistically significant differences between two or more groups of an independent variable on a continuous or ordinal dependent variable. In addition, Spearman correlation coefficient and *p*-values were calculated to assess the relationship between lyso-Gb1 concentrations and disease severity, which was measured by the GD-DS3 scoring system with GraphPad Prism 9. 

## 3. Results

### 3.1. Study Cohort

The LYSO-PROOF study is an international, multicenter, observational study (ClinicalTrials.gov registry number: NCT02416661) that has been conducted in Israel, Russia, Pakistan, Egypt, Iran, Morocco, Algeria, India, Spain, Albania, Greece, Sweden, Columbia, and Tunisia. In total, 160 participants were recruited at 22 study sites. Males and females accounted for 49% and 51% of the participants, respectively. The mean age at enrollment was 23 years (SD: 17; median 20; range 1–81) for all participants (Table 1). 

### 3.2. Genetic Characterization of the Cohort

Although genotype–phenotype correlations in GD have limitations and a significant overlap in clinical manifestations is found between individuals with GD1, 2, and 3 [2], the following rules were applied for the classification into GD1 and primary neurologic disease (nGD). Individuals with at least one p.Asn409Ser allele and individuals who are homozygous for the p.Asn409Ser, p.Leu393Val, p.Arg398Gln, and p.Arg87Trp variants do not develop primary neurologic disease and are classified as GD1 (*n* = 114) [2,25,26]. Individuals who are homozygous for the p.Leu483Pro, p.Asp448His, p.Phe252Ile, and p.Leu279Val variants and individuals with the variant p.Leu483Pro in combination with p.Gly416Ser, p.Asp448His, p.Leu279Val, or complex *GBA1* rearrangements typically present with severe disease, often associated with neurologic complications, and are therefore excluded from the study [2,25,27]. Forty-six participants could not be categorized due to (i) having previously unreported genotypes; and (ii) not fulfilling the criteria to be classified as nGD at the time of diagnosis, and are therefore listed as not classified (Table 1, Figure 1, and Appendix A). 

Sequencing of the *GBA1* gene revealed a total of 44 distinct disease-causing variants on the 320 *GBA1* alleles of the 160 study participants (Appendix A). By far, the most frequent variant was c.1226A > G p.(Asn409Ser), accounting for 54.7% (*n* = 175) of all alleles. Figure 1a provides an overview of variants with repeated observations (*n* ≥ 4) in the entire cohort and a comprehensive list of all disease-causing variants detected is provided in Appendix A.

The 160 participants had 48 distinct *GBA1* genotypes (Appendix A). The three most frequent genotypes were (i) homozygosity for c.1226A > G p.(Asn409Ser) (*n* = 66; 41.2%); (ii) compound heterozygosity for c.1448T > C p.(Leu483Pro)/c.1226A > G p.(Asn409Ser) (*n*=13; 8.1%); and (iii) compound heterozygosity for c.1226A > G p.(Asn409Ser)/c.1342G > C p.(Asp448His) (*n* = 6; 3.8%) (Figure 1b). All other genotypes were each observed in less than 3.5% of the patients. The participants classified as GD1 (*n* = 114) were further subdivided by genotype into mild (c.1226A > G p.(Asn409Ser) homozygous) (*n* = 66) and severe (all other GD1 genotypes) (*n* = 48) (Table 1, Figure 1b, and Appendix A). Figure 1b provides an overview on genotypes and classifications (*n* ≥ 3) and a detailed list of all genotypes is provided in Appendix A.

### 3.3. Distribution of Disease Onset for Several Genotypes

Next, we analyzed disease onset or diagnosis of the indicated cohorts depicted in Figure 2. All patients with GD1 (*n* = 114) had a mean onset of 27 years (median: 25 years; IQR [15–39]), patients with mild GD1 (*n* = 66) had a slightly increased mean onset of 29 years (median: 26 years; IQR [18–39], whereas patients with severe GD1 (*n* = 46) had a mean onset of 26 years (median: 24 years; IQR [8–40]). These results are similar to data collected in the International Collaborative Gaucher Group (ICGG) Gaucher Registry [25].

We further analyzed disease onset for specific subclasses of severe GD1. While the mean age of disease onset in patients with GD carrying p.Asn409Ser/Other (other excludes p.Leu483Pro) (*n* = 30) was 29 years (median: 26 years; IQR [15–42]) and similar to disease onset for patients with mild GD1, disease onset in patients with GD1 carrying p.Asn409Ser/p.Leu483Pro (*n* = 13) have a decreased mean onset of only 24 years (median: 20 years; IQR [7–42]) (Figure 2). 

In contrast, the cohort with GD patients that could not be classified (*n* = 46) due to having previously unreported genotypes had a decreased mean onset of 10 years (median: 6 years; IQR [2–13]) (Figure 2). 

### 3.4. Presence of Clinical Symptoms

The known hallmarks of GD such as organomegaly (spleen and liver), cytopenia, and Gaucher cells were present in the majority of all patients (Table 2, Appendix A) [2]. Splenomegaly (68.8%), thrombocytopenia (74.1%), hepatomegaly (53.6%), anemia (53.1%), and Gaucher cells in bone marrow (71.4%) were present in a large fraction of all patients with GD1, but the presence of these hallmark features was unevenly distributed when comparing the mild and severe GD1 cohorts (Table 2, Appendix A). A comparison for splenomegaly (mild: 54.5% vs. severe: 89.1%), thrombocytopenia (68.2% vs. 82.6%), hepatomegaly (39.4% vs. 73.9%), anemia (39.4% vs. 72.3%), and Gaucher cells in bone marrow (59.4% vs. 83.9%) between mild and severe GD1 patients revealed a striking difference in frequency of presence of GD hallmark clinical symptoms between both cohorts (Table 2). The cohort of GD patients that could not be classified had a frequency of GD hallmark clinical symptoms that was overall slightly increased compared to severe GD1 (Table 2). 

In addition, bone involvement, which is often the most debilitating aspect of GD1 [28], is present in 50.9% of GD1 patients in LYSO-PROOF (mild: 39.4% vs. severe 66.7%). A more detailed list is provided in Appendix A with all other listed clinical symptoms being less frequent.

### 3.5. Lyso-Gb1 Levels in GD Patients

Lyso-Gb1 has been described as a disease-monitoring biomarker for GD [9,17]. A careful analysis yielded a cutoff at 12 ng/mL with an ideal sensitivity and specificity of 100%, which was independent of gender [17]. Lyso-Gb1 values from the first visit of all LYSO-FROOF patients covered almost two orders of magnitude by ranging from 13 to 1520 ng/mL (median: 253 ng/mL; IQR [129–499]). For GD1 patients, the range was from 16 to 911 ng/mL (median: 230 ng/mL; IQR [119–400]) (Figure 3 and Table 3). 

### 3.6. Correlation of lyso-Gb1 with Disease Severity (GD-DS3)

The disease severity scoring system (DS3) is an established and validated measure for evaluating GD1 severity based on bone, hematologic, and visceral domains [5,6,7]. Disease severity states are defined as mild (DS3 < 3.0), moderate (DS3 3.0–6.0), marked (DS3 6.0–9.0), and severe (DS3 9.0–19.0) [5]. Overall, the GD-DS3 score could be assessed for 60% of all patients (96/160) (Table 3). All patients with GD1 (*n* = 66) had a mean DS3 of 1.67 (median: 1.4; IQR [0.7–1.9]). The mild GD1 cohort (*n* = 48) had a mean DS3 of 1.35 (median: 1.4; IQR [0.7–1.7]), compared to a mean DS3 of 2.52 (median: 2.1; IQR [1.3–3.7]) for the severe GD1 cohort (*n* = 18), and a mean DS3 of 3.81 (median: 3.6; IQR [2.4–5.2]) for the not classified cohort (Table 3). 

Finally, we investigated a potential correlation between lyso-Gb1 concentrations and disease severity measured by the DS3 scoring system in all patients with GD. 

Overall, there was a moderate, statistically highly significant correlation between lyso-Gb1 levels and disease severity measured by the DS-3 scoring system in all patients (*n* = 96) (r = 0.602; *p* < 0.0001) (Figure 4). All patients with the GD-DS3 score available (mild GD1 (*n* = 48), severe GD1 (*n* = 18), and not classified GD (*n* = 30)) were included in the Spearman correlation analysis (Figure 4). 

## 4. Discussion

With 160 newly diagnosed, genetically confirmed patients with GD, who have never received disease-specific treatment, the LYSO-PROOF study is one of the biggest of its type. This GD cohort captures not only the phenotype, but also covers many geographical areas.

In total, the enrolled patients had 48 distinct *GBA1* genotypes, and these were stratified into GD1 (*n* = 114; 71.3%), and further subdivided dependent on the genotype in mild GD1 (*n* = 66; 41.3%) and severe GD1 (*n* = 48; 30.0%). Forty-six patients had a genotype that could not be strictly attributed to GD1 or nGD due to lacking information for the *GBA1* variants of these patients. All patients for which no established classifier GBA1 genotype could be determined were combined under not classified (*n* = 46; 28.8%) (Table 1, Figure 1, and Appendix A). 

Whereas neuropathic GD patients are generally associated with a severe phenotype, present with early disease onset and characterized by presence of primary neurological symptoms, GD1 patients are characterized by absence of primary neurological symptoms, generally develop milder phenotypes, and are therefore coming to clinical attention at a much older age [25]. The data from the LYSO-PROOF study demonstrate that although hallmarks of GD such as organomegaly (spleen and liver), anemia, thrombocytopenia, and bone involvement were present in most GD1 patients, the presence and combination of these hallmark features were heightened in severe compared to mild GD1 patients—indicating that classification based on genotypes is a useful tool to predict disease severity in GD1 patients (Table 2 and Appendix A). 

A very wide variation in disease onset was observed within all GD1 patients (range: 2–81 years) (Table 1). It is estimated that up to 50% of p.Asn409Ser homozygotes may never come to medical attention, due to the absence of noticeable clinical manifestations [29]. While the median onset for all GD1 patients was 25 years, disease onset in GD patients with p.Asn409Ser/p.Leu483Pro has a decreased median onset of 20 years (Figure 2). In contrast, the cohort with GD patients that could not be classified due to having previously unreported genotypes had a median onset of only 6 years. As these GD patients are coming to clinical attention at an earlier stage, they are most likely affected with a more severe phenotype including GD2 and GD3, respectively.

Lyso-Gb1 is generated as a consequence of diminished glucocerebrosidase activity and is regarded as the most sensitive and GD-specific biomarker [9], with a cutoff at 12 ng/mL [17]. Lyso-Gb1 levels at the point of enrollment in LYSO-PROOF ranged from 13 to 1520 ng/ml, with a median of 230 ng/ml for all GD1, 167 ng/ml for mild GD1, 320 ng/mL for severe GD1 patients, and 442 ng/ml for the cohort with GD patients that could not be classified (Table 3). The differences in lyso-Gb1 values for mild GD1 compared to a) severe GD1 and b) not classified cohort were both highly significant (*p* < 0.0001), while the comparison between severe GD1 and not classified cohorts was not significant (Figure 3). 

Disease severity of GD patients in the LYSO-PROOF study was assessed with the validated disease severity scoring system (GD-DS3), which is based on bone, hematologic, and visceral domains [5,6]. All patients with the GD-DS3 score available (*n* = 96) were included to investigate a potential correlation between lyso-Gb1 concentrations and disease severity. Overall, there was a moderate, statistically highly significant correlation between lyso-Gb1 levels and disease severity measured by the DS-3 scoring system in all patients (r = 0.602; *p* < 0.0001) (Figure 4). 

## 5. Conclusions

Using medical, genetic, and biochemical data from the 160 newly diagnosed GD patients in the LYSO-PROOF study, we aimed to investigate the application of lyso-Gb1 as a predictive parameter for the phenotypic severity of the patient’s genotype. Though lyso-Gb1 levels were widely distributed in our cohort, and they displayed a moderate and statistically highly significant correlation with disease severity assessed by the GD-DS3 scoring system. From a clinical point of view, the results indicate that even for patients with unclear clinical status and genotype, lyso-Gb1 levels could predict disease severity in GD patients. Furthermore, it could be used as a valuable addition in newborn screening programs to shorten the diagnostic delay in GD. In conclusion, our findings support the utility of lyso-Gb1 as a sensitive biomarker for GD and indicate that it could help to predict the clinical course of patients with undescribed genotypes to improve personalized care in the future. 

## Figures and Tables

**Figure 1 diagnostics-13-02812-f001:**
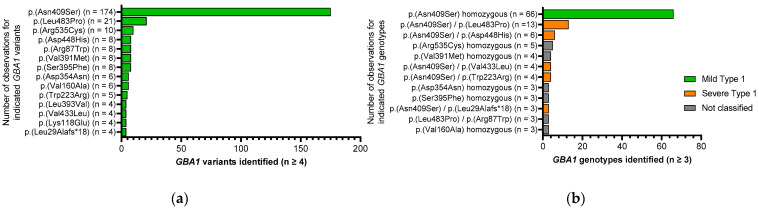
The most frequent *GBA1* variants and genotypes identified in the LYSO-PROOF study. (**a**) *GBA1* variants (*n* ≥ 4) identified. (Analyses based on information from 320 variants in 160 patients). (**b**) *GBA1* genotypes (*n* ≥ 3) identified. Genotypes for mild type 1, severe type 1, and not classified are represented in the indicated colors.

**Figure 2 diagnostics-13-02812-f002:**
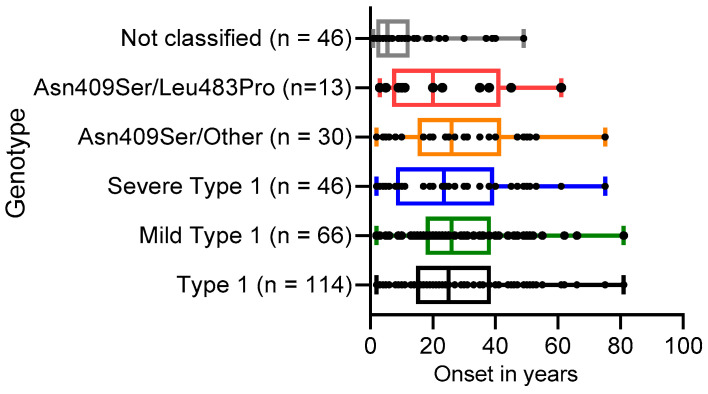
Distribution of onset for several genotypes in GD. Onset represents the age of diagnosis or recognition. The boxes represent the median and upper and lower quartiles for the indicated genotypes. Symbols represent the data from individual patients for the genotypic groups. The cohort p.Asn409Ser/Other excludes the variant p.Leu483Pro.

**Figure 3 diagnostics-13-02812-f003:**
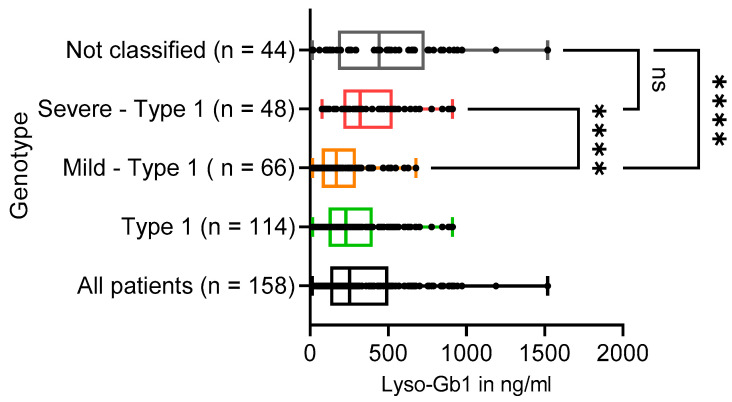
Lyso-Gb1 levels in GD patients. Lyso-Gb1 levels in ng/mL are shown for the indicated genotypes. The boxes represent the median and upper and lower quartiles for the indicated genotypes. Next, we compared with each other the subcohorts of mild GD1, severe GD1, and the not classified GD cohort. While the mild GD1 cohort showed a median lyso-Gb1 concentration of 167 ng/mL (IQR [74–293]), the severe GD1 cohort had a median of 320 ng/mL (IQR [213–528]), and the not classified cohort had a median of 442 ng/mL (IQR [178–732]) (Figure 3 and Table 3). Differences in lyso-Gb1 values for mild GD1 compared to (a) severe GD1 and (b) not classified cohort were both significant (**** *p* < 0.0001), while the comparison between severe GD1 and not classified cohorts was not significant (ns) (Figure 3).

**Figure 4 diagnostics-13-02812-f004:**
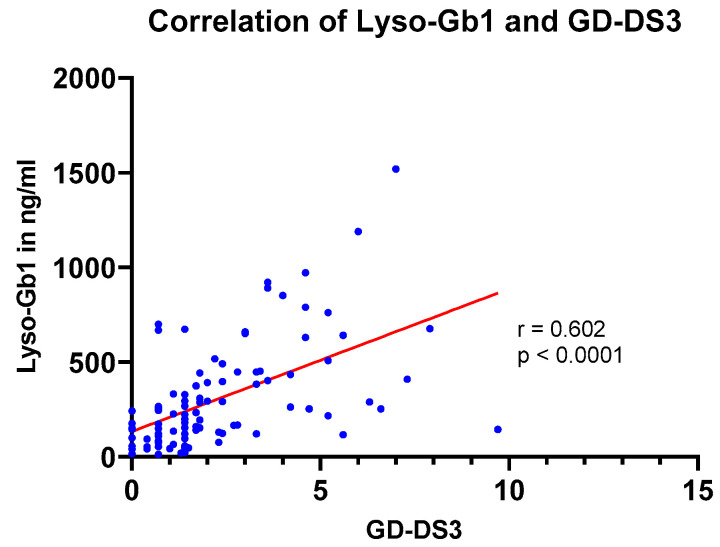
Spearman correlations between lyso-Gb1 levels and disease severity (GD-DS3) for all patients (*n* = 96).

**Table 1 diagnostics-13-02812-t001:** Characteristics of participants with GD.

Parameter	All Patients *N* = 160	Type 1 *N* = 114	Mild—Type 1 *N* = 66	Severe—Type 1 *N* = 48	Not Classified *N* = 46
Gender, n (%)					
Male	78 (49%)	61 (51%)	29 (44%)	30 (62%)	19 (41%)
Female	82 (51%)	58 (49%)	37 (56%)	18 (38%)	27 (59%)
Age at diagnosis (years)					
Mean (SD)	23 (17)	27 (17)	29 (16)	26 (18)	10 (12)
Median [IQR] [range]	20 [6–35] [1–81]	25 [15–39] [2–81]	26 [18–39] [2–81]	24 [8–40] [2–75]	6 [2–13] [1–49]
Distribution by country, *n*					
Israel (2)	55	53	46	7	2
Russia (1)	26	25	10	15	1
Pakistan (1)	24	1	-	1	23
Egypt (1)	12	7	1	6	5
Iran (1)	12	6	3	3	6
Morocco (2)	9	7	1	6	2
Algeria (4)	8	5	3	2	3
India (3)	4	-	-	-	4
Spain (2)	3	3	-	3	-
Albania (1)	3	3	-	3	-
Greece (1)	1	1	-	1	-
Sweden (1)	1	1	1	-	-
Colombia (1)	1	1	1	-	-
Tunisia (1)	1	1	-	1	-

**Table 2 diagnostics-13-02812-t002:** Frequency of presence of clinical symptoms.

		Condition	Present		
Clinical Symptoms	All Patients *N* = 160	Type 1 *N* = 114	Mild Type 1 *N* = 66	Severe Type 1 *N* = 48	Not Classified *N* = 46
	(%) *N* present/*N* total	(%) *N* present/*N* total	(%) *N* present/*N* total	(%) *N* present/*N* total	(%) *N* present/*N* total
Splenomegaly	(75.0) 117/156	(68.8) 77/112	(54.5) 36/66	(89.1) 41/46	(90.9) 40/44
Thrombocytopenia	(74.4) 116/156	(74.1) 83/112	(68.2) 45/66	(82.6) 38/46	(75.0) 33/44
Hepatomegaly	(62.4) 98/157	(53.6) 60/112	(39.4) 26/66	(73.9) 34/46	(84.4) 38/45
Anemia	(56.3) 89/158	(53.1) 60/113	(39.4) 26/66	(72.3) 34/47	(64.4) 29/45
Gaucher cells in bone marrow	(75.8) 75/99	(71.4) 45/63	(59.4) 19/32	(83.9) 26/31	(83.3) 30/36
Bone involvement	(43.1) 69/160	(50.9) 58/114	(39.4) 26/66	(66.7) 32/48	(23.9) 11/46

Bone involvement = presence of one of the following clinical symptoms: bone pain, osteolytic lesions, osteonecrosis, pathologic fractures, bone crises, vertebral compression, avascular necrosis of femoral head, or Erlenmeyer flask deformity of the femurs.

**Table 3 diagnostics-13-02812-t003:** GD-DS3 score and lyso-Gb1 values in GD patients.

	All Patients *N* = 160	Type 1 *N* = 114	Mild—Type 1 *N* = 66	Severe—Type 1 *N* = 48	Not Classified *N* = 46
**GD-DS3**	*N* = 96	*N* = 66	*N* = 48	*N* = 18	*N* = 30
Mean (SD)	2.34 (2.0)	1.67 (1.5)	1.35 (1.3)	2.52 (1.7)	3.81 (2.2)
Median [IQR] [range]	1.7 [0.7–3.3] [0–9.7]	1.4 [0.7–1.9] [0–7.9]	1.4 [0.7–1.7] [0–7.9]	2.1 [1.3–3.7] [0–5.6]	3.6 [2.4–5.2] [0–9.7]
**Lyso-Gb1 in ng/mL**	*N* = 158	*N* = 114	*N* = 66	*N* = 48	*N* = 44
Mean (SD)	337.5 (270.2)	283.2 (213.8)	208.1 (162.1)	386.5 (233.9)	478.2 (344.1)
Median [IQR] [range]	253 [129–499] [13–1520]	230 [119–400] [16–911]	167 [74–293] [16–677]	320 [213–528] [76–911]	442 [178–732] [13–1520]

## Data Availability

The data that support the findings of the study are available on request from the corresponding author, P.B. (email: peter.bauer@centogene.com). The data are not publicly available due to legal constraints.

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
