# Peer review of "Insights into the Value of Lyso-Gb1 as a Predictive Biomarker in Treatment-Naïve Patients with Gaucher Disease Type 1 in the LYSO-PROOF Study"

_diagnostics, 2023, doi:10.3390/diagnostics13172812_

Round 1
Reviewer 1 Report
Dear Authors, congratulations for the paper that appear very interesting in the OMICS era.
The dissertation is fluent and clear by a well conducted study. The concept of personalization of the therapy is essential in the world of rare disorders (medicine of precision). The research of predictive and reliable biomarkers is a frequent unmeet needs in the everyday life. this study could be extended in a larger population. A topic to argue could be the delay in the diagnosis of Gaucher disease and the need to support NBS on this disorder by using Lyso-GB1 as biomarker.
The relationship between Lyso-GB1 and DS-3 scoring is very important. The tables are very clear and readable.
Well written
Author Response
First, we want to thank the reviewer for the positive perception of our manuscript, the encouraging discussion, and valuable comments.
As suggested, we have included a sentence ("Furthermore, it could be used as a valuable addition in Newborn Screening programs to shorten the diagnostic delay in Gaucher disease. ") in the conclusion section (#338) to underline the need of a timely diagnoses in Gaucher disease and the possibility to use lyso-Gb1 in NBS therefor.
Furthermore we had the language of the manuscript checked again by a native speaker who applied minor improvements.
Reviewer 2 Report
Dear Authors,
Gaucher disease is a rare autosomal recessive disorder and Lyso-Gb1 is useful biomarker to monitor the disease progress. This paper is good to show the relationship of disease severity between the disease genotype and the level of Lyso-Gb1. This paper will be suggest to read for the colleagues who worked in the lab of lysosomal storage disorders.
The paper is sound and of importance, and I only have minor comments regarding the data need to be check or there are some typo:
- #227 ... " (Table , Supplementary Table 3)."...
- #266 ..."of all patients (96/166) (Table 3) " ...
Author Response
First, we want to thank the reviewer for the positive perception of our manuscript, the encouraging discussion, and valuable comments. We apologize for the missing or false data, respectively.
As suggested, we have completed the data in brackets in line #227 to (Table 2, Supplementary Table 3) and changed the data in brackets in line #266 to (96/160).